# Co-option of an endogenous retrovirus envelope for host defense in hominid ancestors

Daniel Blanco-Melo[1,2†], Robert J Gifford[3], Paul D Bieniasz[1,2,4*]

[1]Aaron Diamond AIDS Research Center, The Rockefeller University, New York, United States; [2]Laboratory of Retrovirology, The Rockefeller University, New York, United States; [3]MRC-University of Glasgow Centre for Virus Research, Glasgow, United Kingdom; [4]Howard Hughes Medical Institute, The Rockefeller University, New York, United States

**Abstract** Endogenous retroviral sequences provide a molecular fossil record of ancient infections whose analysis might illuminate mechanisms of viral extinction. A close relative of gammaretroviruses, HERV-T, circulated in primates for ~25 million years (MY) before apparent extinction within the past ~8 MY. Construction of a near-complete catalog of HERV-T fossils in primate genomes allowed us to estimate a ~32 MY old ancestral sequence and reconstruct a functional envelope protein (ancHTenv) that could support infection of a pseudotyped modern gammaretrovirus. Using ancHTenv, we identify monocarboxylate transporter-1 (MCT-1) as a receptor used by HERV-T for attachment and infection. A single HERV-T provirus in hominid genomes includes an *env* gene (hsaHTenv) that has been uniquely preserved. This apparently exapted HERV-T *env* could not support virion infection but could block ancHTenv mediated infection, by causing MCT-1 depletion from cell surfaces. Thus, hsaHTenv may have contributed to HERV-T extinction, and could also potentially regulate cellular metabolism.

*For correspondence: pbieniasz@rockefeller.edu

Present address: [†]Department of Microbiology, Icahn School of Medicine at Mount Sinai, New York, United States

Competing interests: The authors declare that no competing interests exist.

## Introduction

When the targets of a given retrovirus include host cells that will become part of the germ-line, viral sequences can become inherited and sometimes fixed in host populations. Indeed, about 8% of the human genome is composed of inactive or fragmented endogenous retroviruses (ERVs) (*Lander et al., 2001*). A few retroviral genes and regulatory DNA elements have been exapted to perform diverse host functions, such as syncytial trophoblast fusion (*Lavialle et al., 2013*), and transcriptional regulation (*Chuong et al., 2016*; *Fort et al., 2014*; *Lu et al., 2014*; *Macfarlan et al., 2012*; *Rebollo et al., 2012*; *Ting et al., 1992*). In particular, a number of endogenous viral elements have been shown to exhibit antiviral activity. Indeed, *gag* (*Best et al., 1996*; *Murcia et al., 2007*; *Yan et al., 2009*), *env* (*Gardner et al., 1991*; *Ito et al., 2013*; *Kozak, 2015*; *McDougall et al., 1994*; *Robinson and Lamoreux, 1976*; *Spencer et al., 2003*), and accessory genes (*Czarneski et al., 2003*; *Frankel et al., 1991*) from endogenous viruses have been shown to contribute to host defense against exogenous retroviruses, through a variety of mechanisms.

Although some functions have been ascribed to exapted retroviral fragments, most provide no obvious advantage to their host. Nonetheless, endogenous proviruses represent a fossil record of past infections, enabling the study of 'paleovirology' (*Emerman and Malik, 2010*). In particular, the synthesis of consensus or deduced ancestral sequences based on retroviral fossils has enabled the functional analyses of reconstructed proteins from ancient, presumptively extinct, retroviruses

**eLife digest** Over millions of years, viruses and the animals that they infect have been locked in a battle for survival, where each has needed to evolve ways to counteract the effects of the other. While the evolution of ancient animals can be studied by looking at the fossilized remains of their extinct relatives, studying how ancient viruses have evolved is more difficult as they usually do not leave behind physical traces of their existence. However, a family of viruses called retroviruses is a notable exception to this rule.

Retroviruses have a step in their life cycle in which their genetic material is integrated into the genome (the name for an organism's complete set of genetic material) of the cell that they have infected. In rare cases, when that cell is a precursor of a sperm or egg cell, then the viral genes may then be passed on to the animal's offspring, ultimately leaving genetic traces that can be studied in modern animals. This acts as a genetic 'fossil record' of extinct viruses.

HERV-T was a retrovirus that spread among our primate ancestors for about 25 million years before its extinction roughly 11 million years ago. Blanco-Melo et al. have now analyzed the genetic remains left by HERV-T in the genomes of humans and related primates, and were able to use this information to recreate a protein that made up the outer envelope that surrounded the virus. Further experiments showed that this viral protein helped HERV-T to infect human cells by interacting with a protein called MCT1 on the cell surface.

Blanco-Melo et al. also found a particular HERV-T gene that was unexpectedly well preserved in the human genome. The gene retained its ability to produce an envelope protein for about 13 to 19 million years. It is likely that ancient primates 'hijacked' the viral gene and used the protein it produced to remove the MCT1 protein from the surface of their own cells. Without MCT1 on the surface, HERV-T was unable to infect the cells. Thus, these findings present an example of how viruses themselves can provide the genetic material that animals use to combat them, potentially leading to their extinction.

(*Dewannieux et al., 2006*; *Goldstone et al., 2010*; *Kaiser et al., 2007*; *Lee and Bieniasz, 2007*; *Perez-Caballero et al., 2008*; *Soll et al., 2010*).

Numerous members of the gammaretrovirus genus currently circulate in mammalian species, but exogenous gammaretroviruses appear absent from modern humans (*Bénit et al., 2001*). A key question in virology, that is potentially answerable using paleovirological analyses, is what caused the elimination of extinct viral lineages? Here, we address this question for an extinct lineage of gammaretroviruses that replicated in ancient primates (HERV-T) (*Blusch et al., 1997*). While no functional HERV-T genes have been found in available modern primate genomes, we show that a single HERV-T *env* gene was apparently exapted in hominids. We find that the product of this HERV-T *env* gene can block retroviral infection mediated by a reconstructed, functional ancestral HERV-T env, by depleting the HERV-T receptor from cell surfaces. Thus, this exapted *env* gene may have driven the extinction of HERV-T from hominids.

## Results

Using similarity search-based approaches, we constructed a comprehensive catalog of HERV-T fossils in old world monkey (OWM) and ape genomes (*Figure 1—source data 1*). Phylogenetic analysis of near-complete proviral elements revealed three major HERV-T clades (T1–T3), each of which was associated with phylogenetically distinct LTR sequences, (*Figure 1—figure supplement 1A*). Two clades (HERV-T3 and HERV-T2) were previously annotated (LTR6A and LTR6B in RepBase [*Bao et al., 2015*]) while a third (HERV-T1) was novel. Other HERV-T-like sequences (found by similarity to HERV-T protein-coding genes) are present in new world monkeys (platyrrhini primates) and documented in Repbase (*Bao et al., 2015*). The presence of orthologous HERV-T loci in various primate species, and integration dates estimated from LTR divergence (*Figure 1—source data 1*), indicated that the infectious ancestor of HERV-T entered primate germlines between ~43 and ~32 MY ago (MYA). Additional germline integration events occurred in various primate lineages for the

ensuing 25 MY, with the most recent integrations that became fixed in primates having occurred ~11 MYA in hominids and ~8 MYA in macaques (*Figure 1—source data 1*).

Comparison of recovered HERV-T fossils indicated that the HERV-T2 clade derives from a single, hypermutated HERV-T1 element (*Figure 1—figure supplement 1B*), and likely expanded by complementation in-trans. HERV-T3 represented the oldest HERV-T clade in catarrhini primates, with integration dates ranging from ~8 to ~30 MYA, and had the highest overall copy number (*Figure 1A* and *Figure 1—source data 1*). There was some support for the division of HERV-T3 into two sub-clades (*Figure 1—figure supplement 1A*). Using a maximum likelihood approach we inferred the sequence of an infectious ancestral HERV-T3 from near complete proviruses (*Figure 1A* and *Figure 1—source data 1*). Unlike fossilized viral elements, the reconstructed ancestral sequence had *gag*, *pol* and *env* genes with full coding potential, and revealed the likely presence of an additional open reading frame (ORF) of unknown function, located 5' to *gag* (*Figure 1B* and *Figure 1—figure supplement 2*). We tested the functionality of the reconstructed ancestral HERV-T envelope gene (ancHTenv,

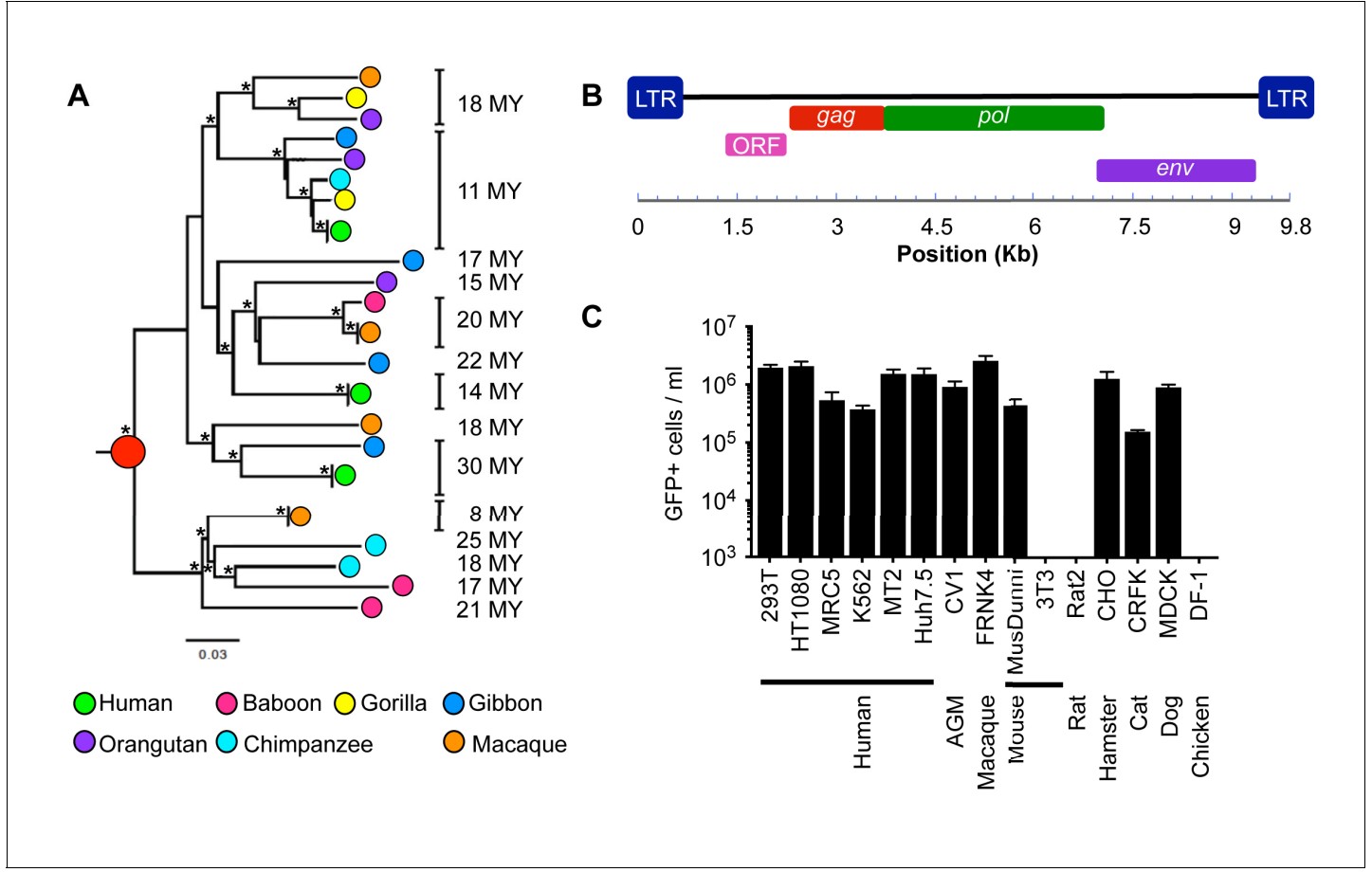

**Figure 1.** Functional reconstruction of a ~32 MY old HERV-T envelope protein. (**A**) Phylogenetic tree of HERV-T3 proviral sequences in OWM and apes. Orthologous sequences are bracketed and estimated integration times indicated. * >90% bootstrap support. Red circle = ancestral root node. (**B**) Ancestral HERV-T3 genome with ORFs indicated. (**C**) Infectivity of MLV particles containing a GFP reporter and pseudotyped with the ancHTenv protein (Mean ± SD, n = 3 replicates). See also *Figure 1—figure supplements 1–2*.

The following source data and figure supplements are available for figure 1:

**Source data 1.** HERV-T elements in OWM and ape genomes.

**Figure supplement 1.** HERV-T proviruses cluster into four monophyletic clades.

**Figure supplement 2.** Deduced sequence of a 32MY old ancestral HERV-T3.

*Supplementary file 1*) by generating murine leukemia virus (MLV) particles containing a *gfp* reporter vector and pseudotyped with ancHTenv. MLV-ancHTenv virions were able to infect many cell lines at high titer ($10^5$–$10^6$ infectious units (IU)/ml, *Figure 1C*) while some rodent (NIH3T3 and Rat2) and chicken (DF-1) cells were comparatively resistant to MLV-ancHTenv (titers $\leq 10^3$ IU/ml).

When challenged with an MLV-ancHTenv expressing a *neo* resistant gene, even DF-1 and NIH3T3 cells could be infected at low efficiency (titers of ~$4 \times 10^2$ and ~$1 \times 10^3$ G418-resistant colonies (G418-RC)/ml). Therefore, DF-1 cells, which exhibited the lowest susceptibility to MLV-ancHTenv, were selected to identify candidate HERV-T receptors as they would give the lowest 'background' level of infection in the context of a cDNA library screen. We generated a lentivirus-based human cDNA library from human 293T cells, which were among the most highly permissive cell lines tested, and then iteratively challenged and selected library-transduced DF-1 cells with MLV-ancHTenv virions

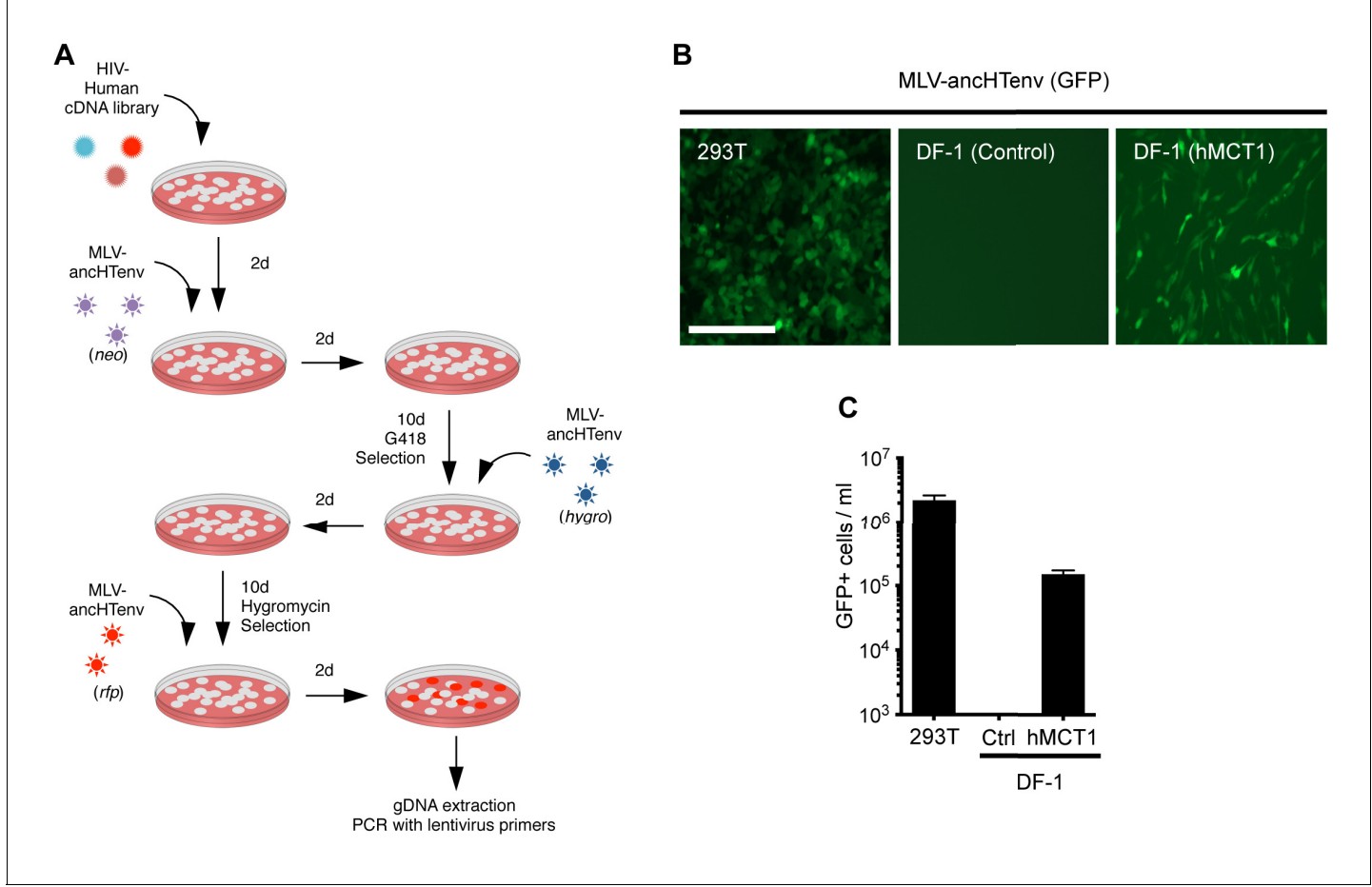

**Figure 2.** MCT-1 functions as a receptor for ancestral HERV-T. (**A**) Scheme of the receptor screening strategy. DF-1 cells were transduced with a lentiviral cDNA library. Two days later, the cells were challenged with MLV-ancHTenv containing a *neo* gene. After a further two days, cells were placed in G418 selection. After another 10 days, G418-resistant cells were replated and challenged with MLV-ancHTenv containing a hygromycin resistance gene. Two days later cells were placed in hygromycin selection. After a further 10 days, Hygromycin-resistant cells were replated and challenged with MLV-ancHTenv containing RFP and were found to be highly susceptible to infection (*Figure 2—figure supplement 1A*). Genomic DNA (gDNA) was extracted from this cell population and subjected to PCR using primers specific to the lentiviral vector (*Figure 2—figure supplement 1B*). (**B**) Fluorescent micrographs of 293T and DF-1 cells, expressing hMCT1 or a control protein, following infection with MLV-ancHTenv expressing GFP as reporter. Scale bar = 200 μm. (**C**) Titers of MLV-ancHTenv/GFP on 293T and DF-1 cells expressing hMCT1 or a control protein (Mean ± SD, n = 2 replicates, one of two separate experiments).

The following figure supplement is available for figure 2:

**Figure supplement 1.** HERV-T receptor identification.

containing antibiotic resistance or fluorescent protein genes (*Figure 2A*). This procedure progressively enriched MLV-ancHTenv-susceptible DF-1 cells (*Figure 2—figure supplement 1A*). Lentiviral-vector directed PCR primers were then used to identify a human cDNA in the selected cells, that encoded human monocarboxylate transporter 1 (hMCT1, also known as SLC16A1, *Figure 2—figure supplement 1B*), a 12-pass transmembrane protein that mediates the transport of monocarboxylates such as lactate and pyruvate across the plasma membrane, and is upregulated in some cancers (*Halestrap, 2012*, *2013*). Reintroduction of an hMCT1 cDNA into naïve DF-1 cells rendered them highly sensitive to infection with MLV-ancHTenv (*Figure 2B and C*). Moreover, MCT1 expression in DF-1 cells also conferred dramatically enhanced ability to bind to MLV Gag-GFP virus-like particles (VLPs) pseudotyped with ancHTenv, but had no effect on the low-level binding of VLPs that lacked an Env protein or incorporated an ecotropic MLV envelope (*Figure 3*). Thus, MCT1 was likely the receptor used by HERV-T to infect ancient old world primates.

Notably, the modern human genome harbors a single HERV-T provirus that includes an *env* gene with nearly full coding potential (*de Parseval et al., 2003*). This *env* ORF lacks only five amino acids at the C-terminus, shares 86% amino acid identity with the reconstructed ancHTenv and is expressed mainly in healthy thyroid tissue (*de Parseval et al., 2003*). The provirus is also present at the orthologous site in gorilla and orangutan genomes, but absent in chimpanzees due to a segmental deletion at that locus. This provirus originated from *bona-fide* retroviral integration (presence of a 4 bp target site duplication) into the internal sequence of another HERV (Repbase: HERVIP10FH), in a locus surrounded by other repetitive sequence elements. Sequence similarity searches using the *env* region suggested the absence of an orthologous insertion in non-hominid primate genomes. Moreover, the divergence between the paired LTRs of this provirus gave an estimated integration date of ~7–19 MYA (*Table 1*). Adjustment of this estimate to account for species distribution leads to the conclusion that this provirus was inserted into the germline of the ancestor of modern hominids at least ~13–19 MYA.

Unlike the reconstructed ancHTenv, the Env protein product of the modern human HERV-T provirus (hsaHTenv) was not able to generate infectious pseudotyped MLV particles (*Figure 4A*). Moreover, unlike ancHTenv-HA, the hsaHTenv-HA protein was not correctly processed and was not incorporated into viral particles (*Figure 4B*). Furthermore, while ancHTenv expression was able to drive syncytium formation, hsaHTenv had no fusogenic activity (*Figure 4C and D*). Inspection of the hsaHTenv and orthologous *env* sequences in gorillas and orangutans revealed mutations at the site where proteolytic cleavage by furin-like proteases generates surface (SU) and transmembrane (TM) subunits (*Hallenberger et al., 1992*) (*Figure 4E*). Insertion of these cleavage site mutations into ancHTenv abolished cleavage and pseudotype infection, while reversion of mutations in hsaHTenv did not correct the cleavage defect and did not yield infectious MLV particles (*Figure 4—figure supplement 1*), Thus, loss of HERV-T envelope function by hsaHTenv was a multi-step process, including loss of furin cleavability, similar to findings with feline endogenous retroviral Env proteins (*Ito et al., 2016*).

Although the hsaHTenv did not maintain function, it has retained a complete open reading frame (ORF) since its deposition in an ancestral hominid genome. To determine the degree to which this ORF preservation should be expected, we deduced an ancestral *env* gene sequence for this particular hominid provirus (ancHTenv[16MYA]) and simulated its neutral evolution using the human neutral substitution rate (*Lander et al., 2001*). Only 6% of sequences maintained the *env* ORF following simulated neutral evolution for 13–19 MY (*Figure 5A*). Note that 6% represents an overestimate as indels were not simulated. Strikingly, inspection of orthologous human and orangutan proviruses revealed highly selective preservation of the *env* ORF, while numerous stop codons (6 to 16) and frameshifts (3 to 5) were present in *gag* and *pol* genes of the same provirus (*Figure 5B* and *Table 2*). For the gorilla provirus, a single nucleotide frameshift was the only lesion in *env*, suggesting recent inactivation. Pairwise sequence comparisons revealed that the human and orangutan *env* genes show signatures of purifying selection (dN/dS ≈ 0.5, p<0.01), whereas comparisons with the gorilla *env* suggest a relaxed selection (*Table 1*). Overall, selective pressures appear to have preserved this HERV-T *env* ORF despite the loss of its retroviral envelope function, and numerous inactivating mutations in the accompanying *gag* and *pol* genes.

We tested the antiviral potential of the hsaHTenv and its direct ancestor in hominids (ancHTenv[16MYA], *Supplementary file 1*) by expressing them into 293T cells, as well as DF-1 clones expressing hMCT1-HA (*Figure 6A*). Strikingly, hsaHTenv and ancHTenv[16MYA] inhibited MLV-

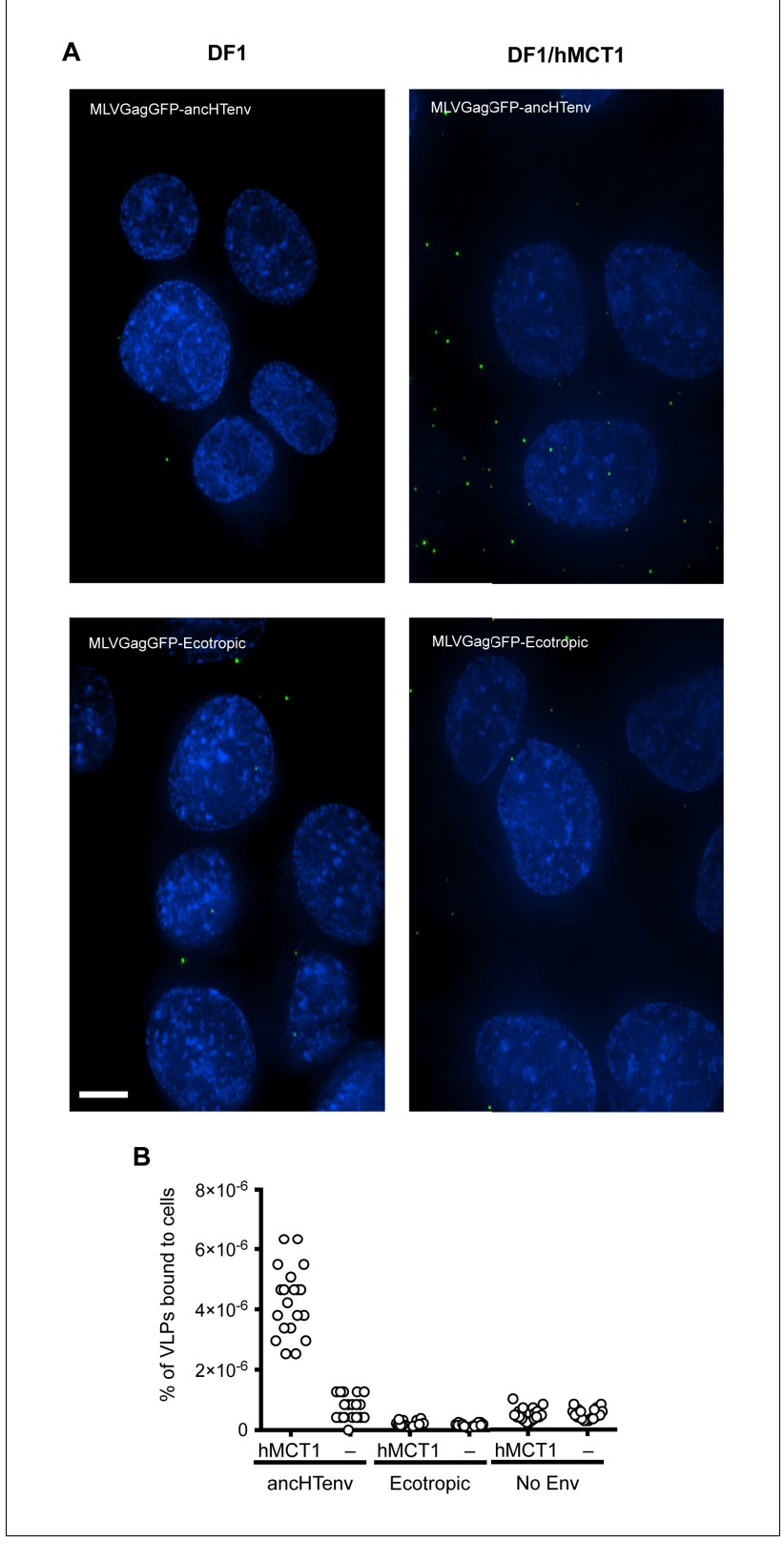

**Figure 3.** Binding of ancHTenv-pseudotyped MLV particles to DF1 cells expressing MCT1. (**A**) Fluorescent micrographs of MLV Gag-GFP VLPs pseudotyped with ancHTenv or ecotropic MLV, bound to DF-1 cells expressing hMCT1 or an empty vector. Scale bar = 5 μm. (**B**) Enumeration of MLV Gag-GFP VLPs bearing the
*Figure 3 continued on next page*

*Figure 3 continued*

indicated Env proteins bound to DF-1 cells expressing MCT-1 or an empty vector. Each data point represents an individual cell (n = 20 for each condition).

ancHTenv infection by >10 fold in both cell lines (*Figure 6B and C* and *Figure 6—figure supplement 1*). Cell clones (293T) stably expressing hsaHTenv were ~50–100-fold less susceptible to MLV-ancHTenv as compared to control cells, but fully susceptible to amphotropic MLV infection (*Figure 6D*). Western blot analyses showed that hsaHTenv, ancHTenv[16MYA] or ancHTenv expression caused depletion of endogenous MCT1 in 293T cells (*Figure 7A*). In DF-1 cells, expression of hsaHTenv-HA, ancHTenv-HA or ancHTenv[16MYA]-HA clearly depleted Flag-tagged hMCT1 from the surface of nearly all cells examined (*Figure 7B* and *Figure 7—figure supplement 1*). In the majority of hsaHTenv-HA, ancHTenv-HA or ancHTenv[16MYA]-HA expressing DF-1 cells, MCT1-FLAG immunofluorescence was nearly undetectable while a few cells showed low levels of intracellular staining (a mixture of diffuse and punctate MCT1-FLAG staining) (*Figure 7—figure supplement 1*). Thus, these data collectively suggest that ancHTenv[16MYA] and hsaHTenv exhibit antiviral activity specifically against HERV-T by causing a reduction in the levels of MCT1 at the cell surface.

## Discussion

These findings suggest a scenario in which HERV-T began to infiltrate primate germ lines, using MCT1 as a receptor ~43–32 MYA. Later, ~19–13 MYA, a HERV-T Env sequence (represented approximately by ancHTenv[16MYA]) was exapted by ancestral hominids and to serve as an antiviral gene. AncHTenv[16MYA] apparently functioned by interacting with MCT1, either at the cell surface promoting its internalization, or in the secretory pathway, blocking its transport to the plasma membrane, leading to MCT1 degradation and its depletion from the cell surface. This process is analogous to the specific receptor interference phenomena observed in retroviral infections (*Nethe et al., 2005*) and exploited by endogenous *env* genes that exhibit antiviral activity in other species (*Gardner et al., 1991*; *Ito et al., 2013*; *Kozak, 2015*; *Robinson and Lamoreux, 1976*; *Spencer et al., 2003*).

Given the fusogenic potential of ancHTenv, it is likely that the progenitor of hsaHTenv would have imposed a fitness cost on its ancestral hominid host. We speculate that the progenitor of hsaHTenv acquired mutations either concurrent with, or after, its integration that resulted in loss

**Table 1.** Molecular evolution analyses of the provirus containing hsaHTenv in its orthologs. (1) Divergence measured in substitutions per site. (2) Integration dates inferred from the divergence of the paired LTRs for human, gorilla and orangutan proviruses. Age is calculated assuming a human neutral substitution rate of $2.2 \times 10^{-9}$ substitutions per site per year. (3) Pairwise dN/dS ratios for the *env* genes calculated using codeml (CodonFreq = F3 × 4, Kappa and Omega estimated). (**) Significantly different from dN/dS = 1 (p<0.01).

| Locus | LTR divergence[1] | Age (MY)[2] | Env dN/dS ratio[3] | |
|---|---|---|---|---|
| | | | Gorilla | Orangutan |
| Human | 0.0317 | 7.20 | 1.4860 | 0.5184** |
| Gorilla | 0.0414 | 9.41 | — | 0.7592 |
| Orangutan | 0.0855 | 19.43 | 0.7592 | — |

**Source data 1.** Likelihood ratio tests on dN/dS estimates for hsaHTenv and its orthologs. dN/dS ratios ($\omega$) were estimated on a pairwise basis using codeml. Likelihood ratio tests were performed comparing the log likelihood of the estimated $\omega$ ($L_1$) to the log likelihood when $\omega$ was fixed to 1 ($L_0$, neutral selection). The probability (P) of twice the difference ($D = 2(L_1 - L_0)$) was calculated using a chi-squared distribution (degrees of freedom = 1). The single nucleotide frame-shift insertion in the gorilla sequence was artificially deleted in order to compare orthologous codons.

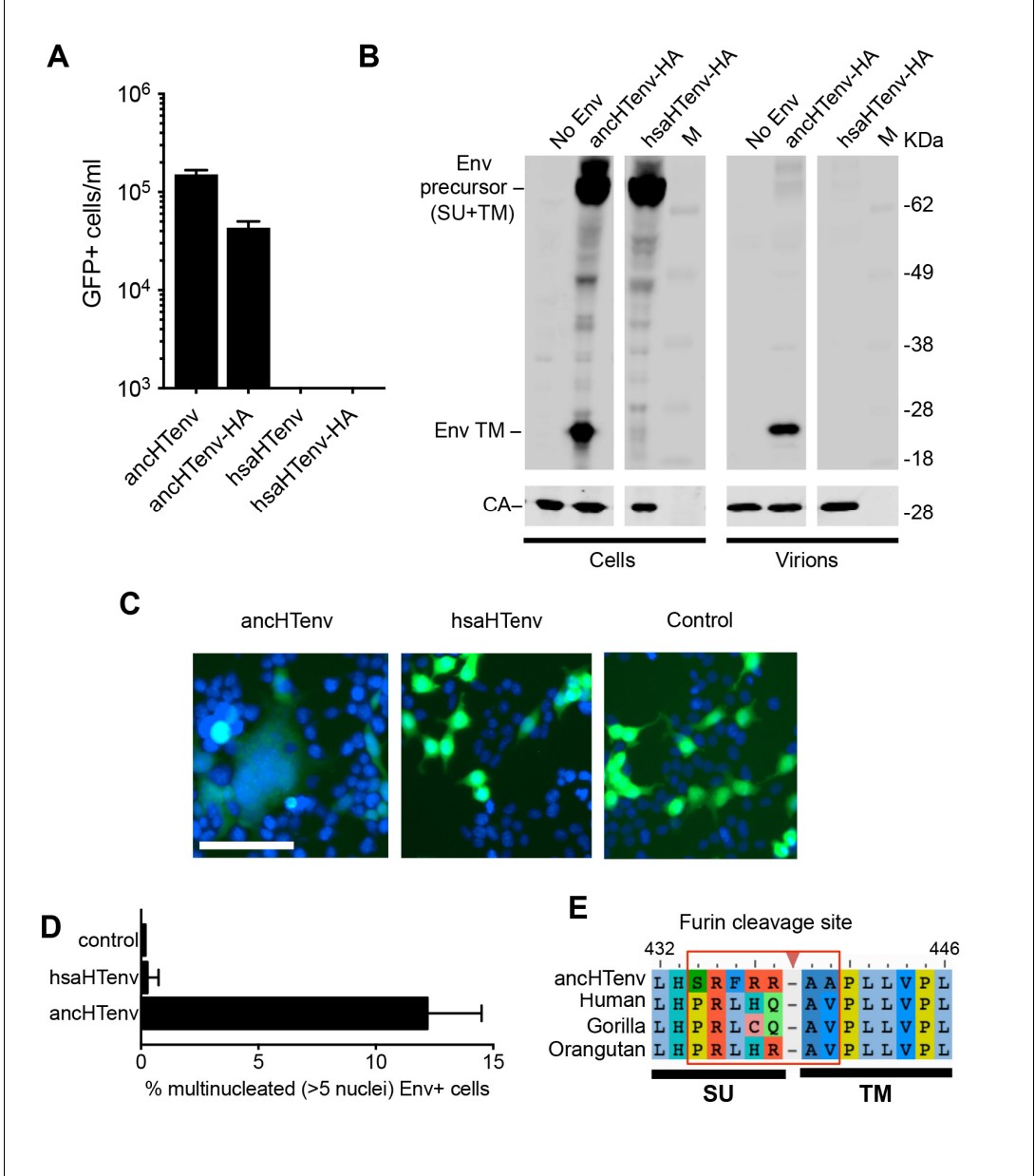

**Figure 4.** The human genome encodes a HERV-T Env ORF that does not function as a retroviral envelope. (**A**) Infectiousness of MLV particles pseudotyped with untagged or C-terminally HA-tagged ancHTenv or hsaHTenv (Mean ± SD, n = 3 replicates, one of two experiments). (**B**) Western blot analyses (α-CA and α-HA) of cell lysates and MLV virions generated following expression of C-terminally HA-tagged ancHTenv or hsaHTenv M: markers. (**C**) Examples of cell fusion in 293T cell cultures expressing ancHTenv or hsaHTenv linked to IRES-GFP. Scale bar = 100 μm. (**D**) Percentage of GFP+ multinucleated cells (>5 nuclei/cell) in 293T cell cultures expressing ancHTenv or hsaHTenv linked to IRES-GFP (Mean ± SD, n = 3 groups of ten microscopic fields, one of two experiments) Blue: DAPI. (**E**) Alignment of ancHTenv and intact or nearly intact HERV-T Env protein sequences encoded by hominid genomes, proximal to the furin cleavage site. See also *Figure 4—figure supplement 1*.

The following figure supplement is available for figure 4:

**Figure supplement 1.** Effects of mutations at the furin cleavage site on ancHTenv and hsaHTenv processing.

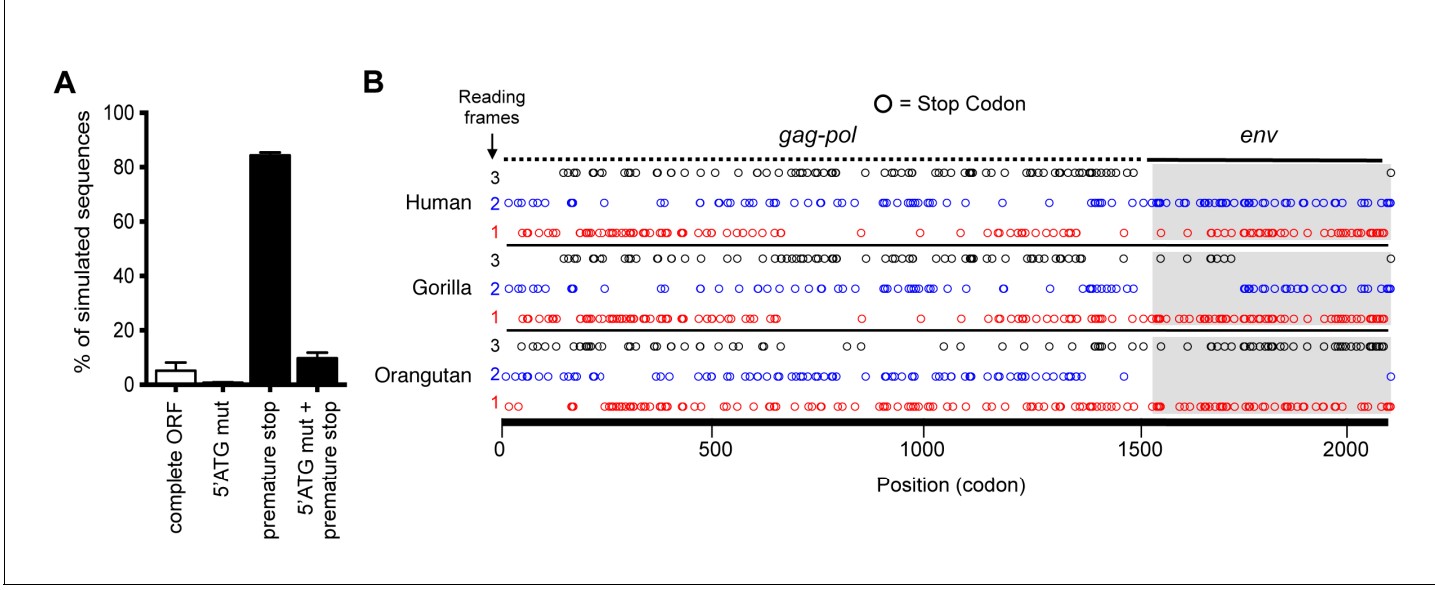

**Figure 5.** Preservation of hsaHTenv and its orthologs in hominids. (A) Monte-Carlo simulations of ancHTenv[16MYA] evolution for 13.45–19.68 MY using a human neutral substitution rate. The percentage of 10,000 simulated sequences of each type is plotted with error bars indicating maximum and minimum estimates. (B) Distribution of stop codons (colored circles) in the Gag, Pol and Env coding sequence of proviruses orthologous to the HERV-T provirus containing hsaHTenv.

of its fusogenic potential (e.g. furin cleavage site mutations). This sequence (represented approximately by ancHTenv[16MYA]) retained full length open reading frame and MCT1 binding activity, and became fixed in the ancestral hominid population. Indeed, it is plausible that this exapted gene might have contributed to the apparent extinction of HERV-T from hominids at some point after the deposition of the most recent fixed hominid germline insertion, ~11 MYA.

The selection pressures acting on hsaHTenv, its orthologs and progenitors might have been complex. Loss of fusogenicity but retention of receptor binding activity are, in a sense, opposing influences in terms of maintenance of the ORF and its function. Moreover, although some purifying selective pressure (low dN/dS) can be detected in hsaHTenv and its orangutan ortholog, a relaxation of those forces might be expected to have occurred after HERV-T extinction, perhaps leading to reduced antiviral activity observed in modern hsaHTenv. Finally, as MCT-1 is a moncaboxylate transporter that is upregulated in human cancers (*Halestrap, 2013*), the ability of hsaHTenv to deplete MCT1 from cell surfaces may suggest an additional or alternative metabolism-related cellular or

**Table 2.** Analysis of inactivating mutations of hsaHTenv encoding provirus in human, gorilla and orangutan genomes. Proviral sequences were aligned to the ancestral HERV-T3 sequence. Nonsense mutations and frameshift indels relative to the ancestral sequence were quantified. 'Indel' = insertion or deletion events compared to the ancestral HERV-T3 sequence that resulted in a reading frame change. 'Stop' = mutations that resulted in a stop codon. (*) Indicates a single nucleotide insertion that results in the truncation of the last five amino acids compared with ancHTenv.

| Locus | Gag (length: 519 codons) | | Pol (length: 1206 codons) | | Env (length: 631 codons) | |
|---|---|---|---|---|---|---|
| | Indel | Stop | Indel | Stop | Indel | Stop |
| Human | 4 | 6 | 3 | 16 | (1*) | 0 |
| Gorilla | 4 | 6 | 4 | 15 | 1 (1*) | 0 |
| Orangutan | 5 | 9 | 3 | 15 | (1*) | 0 |

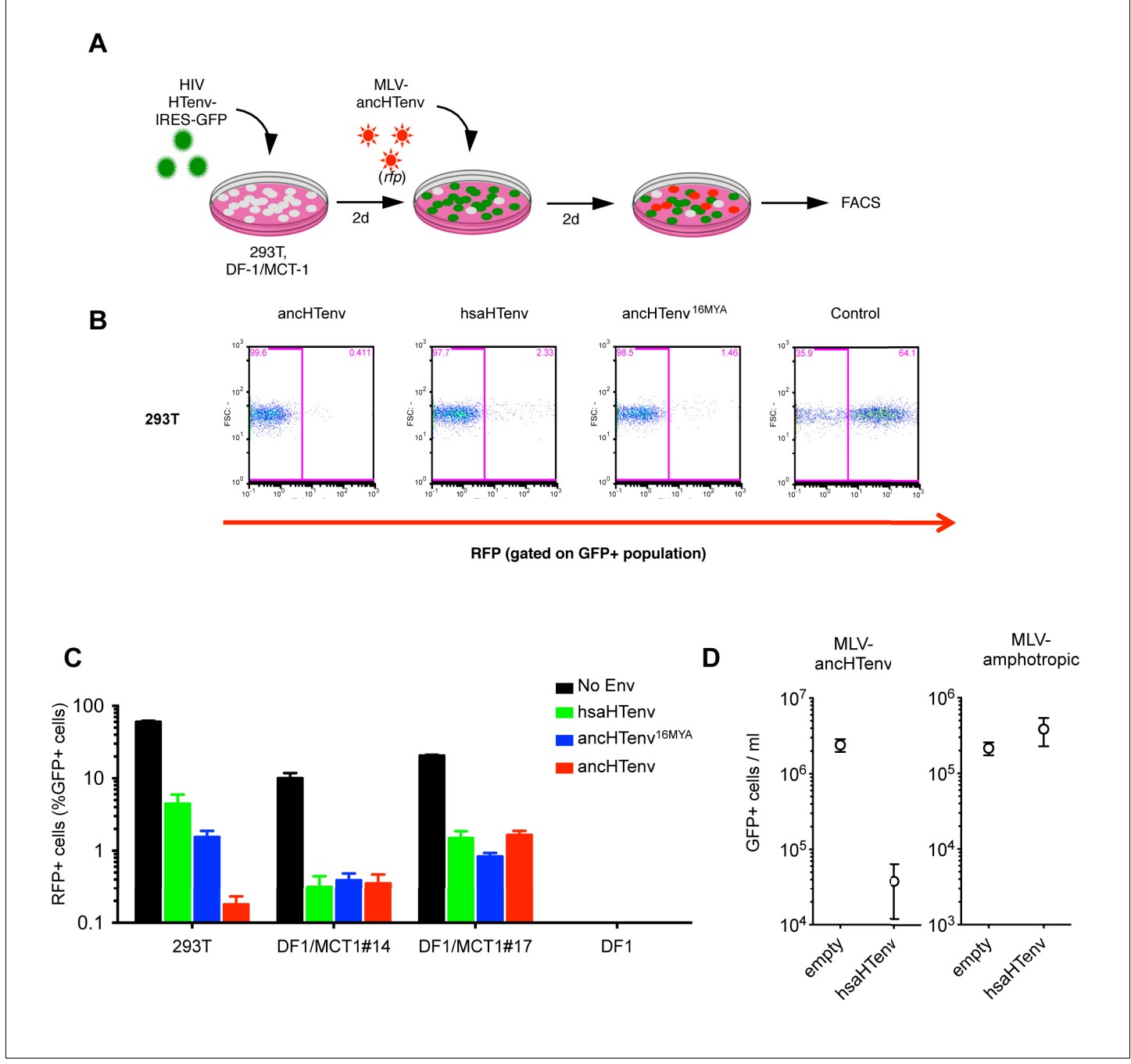

**Figure 6.** The hsaHTenv protein specifically inhibits HERV-T infection. (**A**) Scheme of the antiviral assay. Cells, 293T or DF-1 expressing hMCT1, were transduced with lentiviral vectors expressing HERV-T Env proteins or an unrelated protein (No Env) together with a GFP reporter gene to monitor expression. Cells were challenged with MLV particles pseudotyped with ancHTenv that expresses RFP upon infection. Cell populations were analyzed by FACS 2 days after infection. (**B**) Representative experiment based on the scheme depicted in (**A**) conducted using 293T cells expressing the indicated Env proteins. (**C**) Infectivity of MLV-ancHTenv on 293T cells or two clones of MCT1-expressing DF1 cells (#14, #17), expressing HERV-T envelope proteins according to the scheme described in (**A**). Plots describe the percentage of RFP positive cells (infected) after gating on the GFP positive (Env-expressing) cell population. (Mean ± SD, n = 3 independent experiments). (**D**) Susceptibility of clones of 293T cells expressing an empty vector or hsaHTenv-HA to infection by MLV-ancHTenv/GFP or amphotropic MLV/GFP (Mean ± SD, n = 3 three independent single cell clones assayed once each). See also *Figure 6—figure supplement 1*.

The following figure supplement is available for figure 6:

**Figure supplement 1.** Antiviral activity of HERV-T Env proteins.

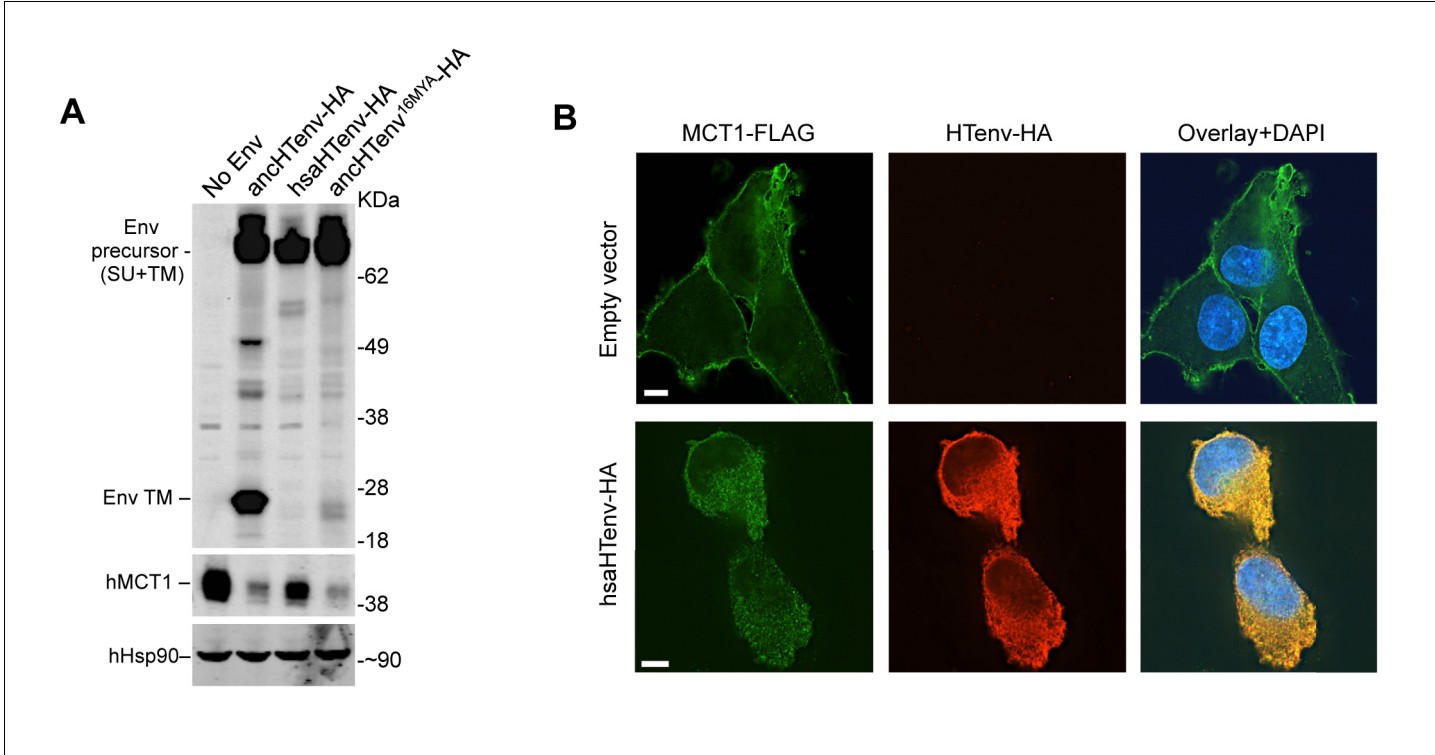

**Figure 7.** The hsaHTenv protein causes depletion of hMCT1 from the cell surface. (**A**) Western blot analyses (α-HA, α-hMCT1 and α-hHsp90) of 293T cell lysates generated following transduction with lentiviral vectors encoding C-terminally HA-tagged HERV-T envelopes. (**B**) Immunofluorescent micrographs of hMCT1-Flag-expressing DF-1 cells transduced with vectors expressing hsaHTenv-HA or an empty vector. Green: α-Flag, red: α-HA, Blue: DAPI. Scale bar = 5 μm. See also *Figure 7—figure supplement 1*.

The following figure supplement is available for figure 7:

**Figure supplement 1.** Additional examples of MCT-1 depletion and relecalization following expression of HERV-T Env proteins.

even anti-tumor functions and selective pressures. Such an activity might have continued to shape hsaHTenv sequence independent of anti-HERV-T activity. In either case, this study highlights the potential importance of ERV proteins as raw material for the innovation of new functions in human ancestors.

# Materials and methods

## Identification of HERV-T elements in primate genomes

Screening for HERV-T elements was performed using a BLAST-based strategy implemented using the Database-Integrated Genome Screening (DIGS) tool (http://giffordlabcvr.github.io/DIGS-tool/). A HERV-T genome consensus sequence (HERV-Tcons, constructed from previously identified HERV-T sequences [*Bénit et al., 2001*]) was used as query to mine old world monkey (OWM) and ape genomes. Specifically, amino-acid and nucleotide sequences of the consensus HERV-T genome were used as probes for tBLASTn (RRID: SCR_011822) (*gag*, *pol* and *env*) and BLASTn (RRID: SCR_001598) (LTR) searches of a genome target database containing complete and low coverage primate genome sequences that were retrieved from publicly available databases: Human (GenBank: GCF_000002125.1), chimpanzee (GenBank: GCF_000001515.6), bonobo (GenBank: GCA_000258655.2), gorilla (GenBank: GCA_000151905.1), orangutan (GenBank: GCF_000001545.4), gibbon (GenBank: GCA_000146795.1), baboon (UCSC: papHam1), and rhesus macaque (GenBank: GCA_000002255.2).

To avoid the identification of sequences from related but distinct retroviruses, significant BLAST hits (e-value < 1E-50) were used as probes for a second round of BLASTx (RRID: SCR_001653) (translated coding hits) or BLASTn (non-coding hits) searches against a sequence reference library containing exogenous and endogenous gammaretroviruses and other class-I ERVs (including HERV-Tcons). Exogenous retroviral sequences were retrieved from the RefSeq database (RRID: SCR_003496) (*Pruitt et al., 2014*), and ERV sequences were based either on consensus sequences of previously published ERV sequence data (*Bénit et al., 2001*; *Sverdlov, 2005*; *Tristem, 2000*; *Villesen et al., 2004*), or previously inferred consensus sequences (*Jern et al., 2005*).

BLAST hits assigned to HERV-T were first ordered by genome scaffold and orientation, and adjacent or overlapping entries were assembled into proviral loci by comparison with HERV-Tcons, allowing for insertions no longer than 10,000 nt.

## Phylogenetic analyses

Due to the high numbers of mutations, especially indels, accumulated in ERV sequences, 44 nearly-complete HERV-T sequences were aligned to HERV-Tcons using MUSCLE (RRID: SCR_011812) (*Edgar, 2004*) in a pair-wise fashion, followed by the creation of a 'gapped' multiple sequence alignment (MSA) using the profile alignment function. Insertions relative to HERV-Tcons were removed from the MSA, but saved in a separate file allowing for the restoration of the removed insertions if required.

Maximum likelihood (ML) phylogenetic trees were constructed from a nucleotide MSA (described above) using raxML (RRID: SCR_006086) (*Stamatakis, 2006*) with the following parameters: rapid bootstrap analysis with 1000 replicates under the GTRCAT model followed by a ML search under GTRGAMMA model to evaluate the final tree topology (-m GTRCAT -# 1000 -x 13 k -f a). Phylogenetic trees were then analyzed using FigTree v1.4.2 (RRID: SCR_008515) (*Rambaut, 2008*). Thereafter, the tree was rooted using a platyrrhini HERV-T-like sequence as outgroup. The resulting tree showed three monophyletic HERV-T clades (corresponding to HERV-T1, HERV-T2 and HERV-T3 in old world primates) whose LTR sequences were clearly distinct. Consensus sequences were derived from each clade and used to re-classify the results from the primate genome screening.

## Molecular evolution analyses of HERV-T proviruses

Dates of integration for HERV-T elements were calculated using PAUP* (RRID: SCR_014931) (*Swofford, 2002*) by determining the divergence (K) from: (i) the corresponding consensus sequence (for soloLTRs), or (ii) to their cognate LTR (for proviral loci flanked by paired LTRs). The resulting K was divided by two times the neutral substitution rate (r) in order to obtain the estimated integration date (K/2r) (*Lebedev et al., 2000*; *Subramanian et al., 2011*). The human neutral substitution rate used was $2.2 \times 10^{-9}$ substitutions per site per year (*Lander et al., 2001*).

APOBEC3-derived hypermutation analysis and statistics were performed using Hypermut 2.0 (RRID: SCR_014933) (*Rose and Korber, 2000*) on 49 *gag-pol-env* HERV-T containing sequences in OWM and ape genomes using default parameters. HERV-T sequences with p-value<0.05 in a Fisher exact-test, were treated as significant for APOBEC3- derived hypermutation.

dN/dS ($\omega$) ratios were calculated on a pairwise basis, by comparing the aligned codon sequences of the human, gorilla (after artificially removing a 1nt frame-shift insertion) and orangutan *env* genes, using codeml from the PAML package of programs (RRID: SCR_014932) (*Yang, 1997*) (runmode = $-2$, CodonFreq = F3 $\times$ 4, Kappa and Omega estimated). Likelihood ratio tests were performed by comparing the log likelihood of the estimated $\omega$ ($L_1$) to the log likelihood when $\omega$ was fixed to 1 ($L_0$, neutral selection). The probability (P) of twice the difference ($D = 2(L_1 - L_0)$) was calculated using a chi-squared distribution (degrees of freedom = 1).

## Ancestral sequence reconstructions

For the ancestral reconstruction of HERV-T3, we first analyzed the 5' and 3' flanking sequences of 22 near complete HERV-T3 proviruses retrieved from old world monkey and ape genomes. These analyses resulted in fourteen unique HERV-T3 integration events (orthologous groups). Ancestral nodes for each orthologous group were used for the ancestral reconstruction.

Initial ML ancestral reconstructions were guided by a MSA together with a phylogenetic tree using baseml (PAML (RRID: SCR_014932) (*Yang, 1997*), model: REV, branch lengths were used as

initial values). Initial values of alpha and kappa were calculated on the MSA by jmodeltest (*Darriba et al., 2012*). A correction for the effect of methylation-induced mutations at CpG dinucleotides was applied on both strands of all ancestral reconstructed sequences as previously described (*Goldstone et al., 2010*).

The resulting ancestral HERV-T3 sequence was further refined in two locations. A single nucleotide insertion, relative to the ancestral HERV-T1 sequence, was eliminated in the 5' pre-*gag* ORF resulting in the suppression of a frame-shift mutation and an expansion of 43 codons relative to the original ancestral HERV-T3 sequence. Additionally, the state of particularly polymorphic si tes in the pre-*gag* ORF was hand curated by a combination of their frequency and the phylogenetic relationships between the corresponding HERV-T3 sequences. The variation present in HERV-T1 sequences was used to break possible ties. A similar procedure was performed on the initial HERV-T3 ancestral *env* sequence with the additional consideration that if the CpG reversion procedure, at a particular position, resulted in the change in chemical nature of the amino acid, the residue at this position was reverted to its 'pre-CpG reversion' state. The final revised sequence corresponded to ancHTenv (*Supplementary file 1*).

The ancestral reconstruction for ancHTenv[16MYA] (*Supplementary file 1*) was performed manually by selecting residues based on the phylogenetic relationships of the three Env sequences present in human, gorilla and orangutan. Polymorphic sites were resolved by comparison with the ancHTenv sequence.

## Identification of hsaHTenv orthologs

Characterization of the hsaHTenv orthologs in other primate species and the ~196 Kb segmental deletion in the Chimpanzee genome, was done using the UCSC genome browser (RRID: SCR_005780) (*Kent et al., 2002*) and tools implemented therein. Stop codon analysis was performed by translating the proviral sequences in all three forward reading frames and the location of their stop codons was plotted using R (RRID: SCR_001905). Frameshift mutations were identified by aligning proviral sequences to the ancestral HERV-T3 sequence.

## In silico simulation of neutral evolution

Monte-Carlo simulations of in silico neutral evolution on the ancHTenv[16MYA] were performed using seq-gen (RRID: SCR_014934) (*Rambaut and Grassly, 1997*) under the GTR model (10,000 iterations) as previously described (*Katzourakis and Gifford, 2010*). Expected branch lengths were calculated for the minimum and maximum estimates of the origin of hominids (13.45 and 19.68 MY respectively) (*Perelman et al., 2011*) using the neutral substitution for humans. The simulated 10,000 sequences were then evaluated for the presence of a 5' methionine and premature stop codons.

## Analysis of HERV-T Env protein sequences

Analysis for the features of HERV-T envelope sequences was performed using tmhmm 2.0 (RRID: SCR_014935) (*Krogh et al., 2001*) for transmembrane and hydrophobic domains, and ProP1.0 (RRID: SCR_014936) (*Duckert et al., 2004*) for signal peptide and propeptide cleavage sites.

## Plasmid construction

The codon-optimized sequences for expression in human cells of ancHTenv, hsaHTenv (GenBank: AB266802) and ancHTenv[16MYA] were synthesized (Genewiz, South Plainfield, NJ) and subsequently inserted into the pCAGGS expression vector (*Niwa et al., 1991*) using EcoRI and XhoI (NEB, Ipswich, MA) restriction enzymes.

Furin cleavage site modified HERV-T envelopes were generated by exchanging the furin cleaveage sites of ancHTenv and hsaHTenv, and vice versa. Specifically, mutagenic PCR primers that annealed to sequences encoding the respective furin cleavage sites were used in overlapping PCR reactions, resulting in the generation of ancHTenv-Furin[Mut] (SRFRRAA to PRLHQAV) or hsaHTenv-Furin[Fix] (PRLHQAV to SRFRRAA). The PCR reactions were treated with DpnI (NEB, Ipswich, MA) restriction enzyme for an hour at 37°C to eliminate plasmid DNA. The complete PCR fragment was inserted into pCAGGS using EcoRI and XhoI restriction sites encoded in the outer primers.

HA-tagged HTenv expression plasmids were generated by introducing two copies of an HA-tag at the C-termini of all HERV-T envelopes using PCR and primers containing the tag DNA sequence

and a 15nt linker sequence. The complete PCR fragments were inserted back into pCAGGS using EcoRI and XhoI restriction sites contained in the PCR primers. HA-tagged HERV-T envelope constructs were also subcloned into pCCIB (a lentiviral expression vector containing a CMV-promoter and IRES-blasticidin resistance cassette), using SfiI (NEB, Ipswich, MA) restriction sites contained in the PCR primers, to generate stably-expressing cell lines and clones. HA-tagged HERV-T envelope constructs or a control protein (GFP) were also subcloned into pCCIGW (lentiviral expression vector containing a CMV-promoter and IRES-GFP), using SnaBI and BstXI (NEB, Ipswich, MA) restriction sites contained in the PCR primers.

The sequence encoding human MCT1 (hMCT1, UniProt: P53985) was amplified from the gDNA of DF-1 cells that had been transduced with a 293T cDNA library and selected as described below. The amplified hMCT1 sequence was inserted into pCCIB utilizing SfiI sites, and the resulting vector used to generate stable DF-1 cell lines expressing hMCT1. C-terminally HA-tagged or Flag-tagged versions of hMCT1 were generated using PCR with primers containing two HA-tag or three Flag-tag encoding sequences and a 15nt linker sequence. The complete PCR fragment was inserted back into pCCIB using SfiI restriction sites contained in the PCR primers.

## Cell culture

All cells were purchased from ATCC (Manassas, VA), except MT2 (RRID: CVCL_2631) and NIH3T3 (RRID: CVCL_0594) that were obtained through the NIH AIDS Reagent Program, and Huh7.5 (RRID: CVCL_7927) that were a gift from Charles M Rice, where they were derived. Cells were assumed to be authenticated by their respective suppliers and were not further characterized. Cells were maintained in Dulbecco's Modified Eagle Medium (DMEM) (293T (RRID: CVCL_0063), DF-1 (RRID: CVCL_0570), HT1080 (RRID: CVCL_0317), K562 (RRID: CVCL_0004), Huh7.5, FRhK4 (RRID: CVCL_4522), MusDunni (RRID: CVCL_9125), NIH3T3 and Rat2 (RRID: CVCL_0513)), Eagle's Minimum Essential Medium (EMEM) (MRC5 (RRID: CVCL_0440), CV-1 (RRID: CVCL_0229), CRFK (RRID: CVCL_2426) and MDCK (RRID: CVCL_0422)), Roswell Park Memorial Institute medium (RPMI) (MT2), or Ham's F-12 media (CHO (RRID: CVCL_0214)) supplemented with 10% FBS or BCS (NIH3T3), 1 mM of L-glutamine (CHO) and gentamycin (2 µg/ml) (Gibco, Waltham, MA) according to ATCC instructions. Mycoplasma testing was not specifically performed, but many cell lines were used in immunofluorescence assays with DAPI staining that should reveal presence of mycoplasma. All cells were incubated at 37°C except DF1 that were incubated at 39°C.

To generate stable cell lines, 293T cells were transfected with plasmids expressing HIV-1 Gag-Pol (pCRV1 [*Zennou et al., 2004*]), VSV glycoprotein and pCCIB plasmids encoding hMCT1 or HA-tagged/untagged versions of HERV-T envelopes using polyethylenimine. In every case virus/vector containing supernatants were harvested and filtered two days after transfection and were used to transduce naïve 293T or DF-1 cells. Transduced cells were expanded in 10 cm dishes with media supplemented with 5 µg/ml (293T) or 20 µg/ml (DF-1) blasticidin S (Thermo Fisher Scientific Inc., Waltham, MA) and were monitored from 3 to 10 days before performing experiments or isolating single cell clones.

## MLV pseudotype infectivity assays

MLV particles pseudotyped with MLV amphotropic (MLV-A) or ecotropic (MLV-E), and different HERV-T envelopes were generated by co-transfecting 293T cells with (i) the corresponding envelope plasmids, plasmids (ii) expressing MLV gag-pol polyprotein (MLVgp) and (iii) an MLV vector encoding a neo gene and/or GFP/RFP (pCNCG/pCNCR) (*Soneoka et al., 1995*) using polyethylenimine. Viruses were harvested and filtered (0.22 µm) two days after transfection. Viral stocks were concentrated using the Amicon Ultra-15 filters (10 kDa) (Millipore, Billerica, MA) before freezing. Cells of interest were infected with serial dilutions of the corresponding virus supplemented with 4 µg/ml of polybrene. Viral titers were calculated by measuring the percentage of infected cells expressing GFP or RFP, 2 days post infection (dpi) using the Guava EasyCyte flow cytometer (Millipore). For MLV particles expressing neo resistance genes, viral titers were calculated by expanding the infected cells in 10 cm dishes with media supplemented with 1 mg/ml (NIH3T3) or 2 mg/ml (DF-1) of G418 and monitoring for 10 days before resistant colonies were counted.

## cDNA library preparation and screening

Total RNA was isolated from a confluent 10 cm dish of 293T cells using Trizol (Invitrogen, Carlsbad, CA). mRNA transcripts were enriched using Oligotex polyA+ resin (Qiagen, Hilden, Germany). Polyadenylated RNA was used to construct a cDNA library using the SMART cDNA Library Construction Kit (Clontech, Mountain View, CA). Briefly, cDNA containing SfiI restriction sites was synthesized using the SMARTScribe MMLV reverse transcriptase (Clontech) with SMART primers. The resulting cDNA was further amplified for 15 cycles using Phusion High-Fidelity DNA Polymerase (Thermo Fisher Scientific Inc.) using SMART primers. PCR products were treated with Proteinase K (Clontech) for 20 min at 45°C before they were digested by SfiI restriction enzyme for 2 hr at 50°C. Digested cDNA was then size fractionated using CHROMA SPIN-400 columns (Clontech). The cDNA containing fractions (>500 bp) were selected to ligate into a pCCIB plasmid encoding corresponding SfiI sites. Overnight ligation was performed using T4 DNA ligase (NEB) and the library was expanded by transformation into electrocompetent DH5α bacteria cells. The 293T cDNA library had a complexity of $3.5 \times 10^6$ colony forming units. Transformed bacteria were cultured as suspended colonies in 6 liters of SeaPrep soft agarose (Lonza, Basel, Switzerland) diluted in 2x LB media at 30°C for 2.5 days. Thereafter plasmid DNA was extracted and used for receptor cloning.

Lentiviral vector stocks carrying the cDNA library were produced by transfecting $6 \times 10^6$ 293T cells with plasmids expressing HIV-1 gag-pol polyprotein (pCRV1), VSV glycoprotein and the cDNA library plasmid (pCCIB). Library containing supernatants were harvested, filtered (0.22 μm), concentrated (Amicon Ultra-15 filters 10 kDa, Millipore) and frozen two days after transfection. DF-1 cells were transduced with the library at an MOI of ~8. Transduced DF-1 cells were then challenged with MLV particles pseudotyped with ancHTenv and containing a *neo* resistance gene (pCNCG) two days later. Infected DF-1 cells were placed in G418 selection at 2 dpi and resistant colonies were collected after 10 days. G418 resistant DF-1 cells were then challenged with ancHTenv pseudotyped MLV particles containing a hygromycin resistance gene (pLHCX). Cells were placed in hygromycin selection at 2 dpi and resistant colonies were collected after 10 days. Hygromycin resistant DF-1 cells were infected with ancHTenv pseudotyped MLV particles expressing RFP (pCNCR) and found to be susceptible to infection. Genomic DNA (gDNA) was extracted from this DF-1 cell population and possible receptor candidates were amplified using PCR and primers directed to the pCCIB vector sequences flanking the SfiI cDNA insertion site. A non-coding isoform of human *atg12* and hMCT1 were amplified from gDNA of transduced and selected DF-1 cells. For all PCRs performed in this study Phusion High-Fidelity DNA Polymerase was used.

## Virus-like particle (VLP) binding assay

Fluorescent MLV VLPs were generated by co-transfecting 293T cells with a plasmid (pCAGGS) expressing MLV Gag-GFP and a plasmid (pCAGGS) expressing ancHTenv, MLV-E or an empty vector, using polyethylenimine. VLPs were harvested and filtered (0.22 μm) two days after transfection and concentrated using the Amicon Ultra-15 filters (10 kDa, Millipore). The VLP binding assay was performed as previously described (*Soll et al., 2010*) using naïve DF-1 cells or a DF-1 clone expressing untagged hMCT1. Fluorescent microscopy images of the cells were acquired using a DeltaVision deconvolution microscope (GE Healthcare, Port Washington, NY). A Z-series of images (capturing the entire thickness of the cell monolayer) were flattened onto a single image and the number of fluorescent particles associated with each cell was counted for 20 distinct cells.

To determine the number of fluorescent VLPs added to the cells, 80 μl of the VLP-containing supernatant was was layered onto 1 ml of 20% sucrose in PBS followed by centrifugation at 20,000 x g for 90 min at 4°C. Pelleted VLPs were resuspended in 300 μl of PBS and filtered (0.22 μm). Two-fold serial dilutions of the VLPs (in PBS) were layered (100 μl) to poly-D-lysine coated glass coverslips and left overnight at 4°C. Fluorescent microscopy images of the VLPs were analyzed using the Delta-Vision software (GE Healthcare) to determine the number of VLPs/ml for each condition. The number of fluorescent particles bound to cells was then calculated as a proportion of the number of fluorescent VLPs added to the cells.

## Western blot assays

Cell lysates and pelleted virions (600 μl of supernatant pelleted through 20% sucrose in PBS) were resuspended in SDS-PAGE loading buffer, with the addition of 0.5% *β*-mercaptoethanol, and

resolved on NuPAGE Novex 4–12% Bis-Tris Mini Gels (Invitrogen) in MOPS running buffer. Proteins were blotted onto nitrocellulose membranes (HyBond, GE-Healthcare) in transfer buffer (25 mM Tris, 192 mM glycine). The blots were then blocked with Odyssey blocking buffer and probed with rat monoclonal anti-MLV capsid (RRID: CVCL_9230, ATCC, R187), rabbit polyclonal anti-HA (RRID: AB_ 217929, Rockland Immunochemicals, Pottstown, PA, 600-401-384), mouse monoclonal anti-HA (RRID: AB_2314672, Covance, Princeton, NJ, MMS-101P), mouse monoclonal anti-hMCT1 (RRID: AB_10841766, Santa Cruz Biotechnology Inc., CA, sc-365501), rabbit polyclonal anti-hHsp90 (RRID: AB_2121191, Santa Cruz Biotechnology Inc., CA, sc-69703) or mouse monoclonal anti-Flag (RRID: AB_262044, Sigma-Aldrich, St. Louis, MO, F1804). The bound primary antibodies were detected using a fluorescently labeled secondary antibody (IRDye 800CW or 680W Goat Anti-Mouse or anti-Rabbit Secondary Antibody, LI-COR Biosciences, Lincoln, NE, RRIDs: AB_621842, AB_621840, AB_10796098, AB_621841). Fluorescent signals were detected using a LI-COR Odyssey scanner (LI-COR Biosciences).

## Immunofluorescence assays

Stable cell populations or single cells clones expressing a particular HA-tagged HERV-T envelope and/or Flag-tagged/untagged hMCT-1 were seeded one day prior to immunofluorescence assay. Cells were fixed with 4% PFA for 30 min followed by treatment with 10 mM glycine (diluted in PBS) for another 30 min. Cells were permeabilized with a PBS containing 0.1% Triton X-100% and 5% goat serum for 15 min. Cells were then washed twice with PBS before being incubated with rabbit anti-HA and mouse anti-Flag antibodies diluted in PBS containing 0.1% Tween-20% and 5% goat serum for 2 hr at room temperature. Cells were washed three times with PBS before being treated with goat anti-mouse and/or anti-rabbit secondary antibody (Alexa Fluor 488 and/or 594 dye, Life technologies, CA, RRIDs: AB_2534091, AB_2534088, AB_2534095, AB_2576217) diluted in PBS containing 0.1% Tween-20% and 5% goat serum for 1 hr at room temperature. DNA was stained with 50 μg/ml of DAPI (diluted in PBS) (Invitrogen) for 5 s. Cells were washed three more times with PBS and fluorescent microscopy images were analyzed using the a DeltaVision microscope (GE Healthcare) or the EVOS FL Cell Imaging System (Thermo Fisher Scientific Inc.).

## Assay for effects of HERV-T Env on infection

Viruses were generated in 293T cells transfected with plasmids expressing HIV-1 gag-pol polyprotein (pCRV1), VSV glycoprotein and pCCIGW (IRES-GFP) plasmids and expressing the various HERV-T envelopes described above, or an unrelated protein as control (SIVgab Nef) using polyethylenimine. Viruses were harvested and filtered (0.22 μm) two days after transfection and were used to transduce $2 \times 10^5$ naïve 293T cells or $1 \times 10^5$ DF-1 clones expressing HA-tagged hMCT1. Transduced cells were expanded in 10 cm dishes until reaching confluency. Afterwards, serial dilutions of MLV expressing RFP (pCNCR) pseudotyped with ancHTenv were used to infect $1 \times 10^4$ transduced cells in 96 well plates. Two days post infection the number of GFP+, RFP+ and double positive cells were counted by FACS using a CyFlow space cytometer (Partec, Görlitz, Germany). The resulting data was analyzed with the FlowJo analysis software. The percentage of RFP positive cells was calculated after gating on the GFP positive population.

## Cell fusion analysis

293T cells were transduced with a lentivirus vector (CCIGW) expressing various HERV-T envelopes or a control protein (SIVgab Nef) (described above) and then fixed and stained with DAPI (described above). Multinucleated cells (>5 nuclei) and cells expressing the GFP reporter (indicative of the expression of the corresponding HERV-T envelope) were counted in three groups of ten microscopic fields and the number of multinucleated GFP+ cells as a percentage of the total number GFP+ cells was calculated for each condition.

## Additional information

### Funding

| Funder | Grant reference number | Author |
|---|---|---|
| National Institute of Allergy and Infectious Diseases | R37AI064003 | Paul D Bieniasz |
| Howard Hughes Medical Institute | | Paul D Bieniasz |
| Medical Research Council | | Robert J Gifford |

The funders had no role in study design, data collection and interpretation, or the decision to submit the work for publication.

### Author contributions

DB-M, Conceptualization, Data curation, Software, Formal analysis, Investigation, Writing—original draft, Writing—review and editing; RJG, Conceptualization, Resources, Data curation, Software, Project administration, Writing—review and editing; PDB, Conceptualization, Formal analysis, Supervision, Funding acquisition, Investigation, Writing—original draft, Project administration, Writing—review and editing

### Author ORCIDs

Robert J Gifford, http://orcid.org/0000-0003-4028-9884
Paul D Bieniasz, http://orcid.org/0000-0002-2368-3719

## Additional files

### Supplementary files

• Supplementary file 1. Sequences of resurrected HERV-T envelopes. Codon optimized (human) ancestral reconstructed sequences for ancHTenv and ancHTenv[16MYA] in Fasta format.

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
