## [Decision Letter]

Thank you for submitting your article "Co-option of an endogenous retrovirus envelope for host defense in hominid ancestors" for consideration by *eLife*. Your article has been favorably evaluated by Wenhui Li (Senior Editor) and three reviewers, one of whom is a member of our Board of Reviewing Editors. The following individual involved in review of your submission has agreed to reveal his identity: Welkin E Johnson (Reviewer #2).

The reviewers have discussed the reviews with one another and the Reviewing Editor has drafted this decision to help you prepare a revised submission.

Summary:

In this manuscript, Blanco-Melo et al. utilize comparative analysis of 'fossilized' endogenous retroviruses of the HERV-T group from human and ape genomes in order to deduce the putative ancestor, and then elegantly demonstrate the ability of its envelope to infect a variety of cell lines when pseudotyped with MLV. The authors identify a candidate human receptor, MCT-1, by challenging very low permissive cells (DF-1) transduced with a cDNA library constructed from highly permissive cells (293T) with ancestral ERV-T env pseudotyped virions; reintroduction of the MCT-1 cDNA alone rendered DF-1 highly permissive to infection mediated by the ancestral envelope. The authors investigate a HERV-T locus that has, quite remarkably, retained its env ORF in humans despite millions of years of evolution, suggesting selective pressures acting on the HERV-T env. Blanco-Melo et al. show that this env is not fusogenic and is not able to infect cell lines that are otherwise sensitive to the ancestral ERV-T env. Endogenous levels of MCT-1 are depleted in the presence of either the HERV-T env or its direct ancestor, and expression of the constructs in MCT-1 expressing cells reduced infection from ancestral ERV-T env pseudo typed MLV, suggesting antiviral protective effects from this exapted viral gene.

The manuscript expands the number of known receptors used by this particular "type" of envelope glycoprotein, which is characterized by covalently linked SU and TM domains, and raises the issue of whether the phenomenon of exaptation of ERV as resistance genes is a widespread phenomenon – although the paper only amounts to proof in principle for the exaptation story, it sets a standard for how Erv glycoproteins should be investigated. It will be interesting to see whether the patterns observed here are unique to HERV-T or represent a predictable pattern among other ERV in other vertebrate lineages. Altogether, this is a very nice study.

The work from Blanco-Melo is compelling and the experimental quality is high. The findings are novel and suggest a scenario of evolutionary preservation of the HERV-T env to confer antiviral protection in primate ancestors following germline invasion of primate ancestors starting ~43-32 mya (and thereby leading to HERV-T extinction), consistent with observations of receptor interference in other retroviruses and exaptation of env for antiviral purposes in other species. The work is very nicely done and concise in presentation and in its findings and interpretation(s) therein. The work clearly offers fundamental biological insight on the basis of the importance and impact of viral proteins in the mechanisms that shape the evolutionary arms race with the host, and the utility of such endogenous sequences as raw material for exaptation in the host. The findings are novel and infer clear biological importance. The conclusions presented by Blanco-Melo et al. are clear and are justified by the experimentation. Likewise, the experiments are designed succinctly to directly test and in turn support the authors' overall hypothesis. In my opinion, there are not major concerns that should impede publication of the work, although minor concerns are pointed out below.

Essential revisions:

1) The authors estimate HERV-T to have begun germline invasion ~43-32 mya, and estimate the HERV-T3 clade insertions, as a group, from ~8-30 mya; the insertion containing the preserved env reading frame was placed at ~7-19 mya, and then adjusted based on species distribution. The age estimation for the 'most recent' hominid insertion is then given in the Discussion as ~11mya. Is this value as obtained for the adjusted estimation for the insertion? Is this meant to suggest that insertion was the last ERV-T to enter the hominid line?

2) In Figure 4 the authors show depletion of endogenous MCT-1 in 293T cells in the presence of the ancestral ERV-T and HERV-T envelope proteins. I am curious about the relative levels of depletion between the various envelope proteins. MCT-1 appears to be comparatively less in the presence of the ancestral envelopes than in the hsaHTenv; This would imply the contemporary human hsaT env has, over time, lost at least some MCT-1 interference. In terms of the evolution of the envelope protein and receptor does this suggest the ancestral forms are 'better' at targeting MCT-1?

3) The experiments showing that HERV-T envelope expression can block infection are convincing, but I have some concerns with the evidence that the hsaHTenv ORF was maintained due to selective pressure. The evolution simulation results are interesting, but not very informative without knowing the number of HERV-T proviruses in the genome, e.g. if there were 20 HERV-T env ORFs that integrated during that time period, on average we would expect one ORF to be maintained by chance, according to this simulation. The analysis is further complicated by the presence of older proviruses that would have had a smaller, but non-zero probability of having their env ORFs preserved to the present day by chance. It would be helpful to have information on dN/dS ratios for these sequences – if this gene has been exapted some level of purifying selection would be expected. If its only function was to protect against a long-extinct virus the selective pressures would have been relaxed since then, which might obscure the signs of selection somewhat, but it seems likely that at least some signature of selection at the residue level would remain.

4) For reasons that are unclear to us, this paper was submitted as a Short Report, although it contains much more than the "single series" of experiments specified for this format. The result is that some results that could be included in the results are relegated to supplemental, and both the Introduction and Discussion are severely constrained. There is a reasonably rich body of literature on exaptation of endogenous viruses that prevent infection by exogenous retroviruses that is barely mentioned here. These include not only the ALV and MLV case briefly mentioned in the Discussion, which, like HERV-T encode blocking env proteins, but also MMTV and JSRV which use very different mechanisms. Also, worth discussing is the possible contribution of the furin cleavage mutations or the C-terminal deletion in exaptation. The reader is shortchanged by the brevity enforced by the format.

---

## [Author Response]

*Essential revisions:*

*1) The authors estimate HERV-T to have begun germline invasion ~43-32 mya, and estimate the HERV-T3 clade insertions, as a group, from ~8-30 mya; the insertion containing the preserved env reading frame was placed at ~7-19 mya, and then adjusted based on species distribution. The age estimation for the 'most recent' hominid insertion is then given in the Discussion as ~11mya. Is this value as obtained for the adjusted estimation for the insertion?*

No, the ~11 MYA age estimate is based on the divergence of paired LTRs in near complete fossilized proviruses (Figure 1 and [Supplementary-material SD1-data]). These data suggest that the most recent insertions occurred about 11MYA, with one noticeable outlier. This is indicated at the end of the first paragraph of the Results section, “with the last detectable integration having occurred ~8 MYA” (Figure 1). This outlier is found only in macaques, not hominids.” We have changed the wording at the end of the first paragraph of the Results section, and in the Discussion to be clearer on this point.

*Is this meant to suggest that insertion was the last ERV-T to enter the hominid line?*

We did not intend to suggest that the 11MYA and 8MYA integrations are the last to have occurred in primates, however we concede that the wording in the original draft did suggest this. We can reasonably state that 11MYA and 8MYA are estimates for the dates at which the most recent integrations *that became fixed* in ancestral primates occurred. It is possible that additional more recent integrations occurred that never became fixed and were purged from primates, and are thus invisible to us. Again, we have amended the wording in the first Results paragraph and the Discussion to be more explicit about this.

*2) In Figure 4 the authors show depletion of endogenous MCT-1 in 293T cells in the presence of the ancestral ERV-T and HERV-T envelope proteins. I am curious about the relative levels of depletion between the various envelope proteins. MCT-1 appears to be comparatively less in the presence of the ancestral envelopes than in the hsaHTenv; This would imply the contemporary human hsaT env has, over time, lost at least some MCT-1 interference. In terms of the evolution of the envelope protein and receptor does this suggest the ancestral forms are 'better' at targeting MCT-1?*

We agree that the absence of recent selection pressure on hsaHTenv (after the “extinction” of HERV-T) and/or a possible additional neofuctionalization to perform a distinct but related function (e.g. metabolic modulation) might result in hsaHTenv being less potent at MCT1 downregulation than the ancestral reconstructed proteins. However, our data is actually a little ambiguous on this point – the ancestral forms appear more potent antiviral proteins than hsaHTenv in 293T cells, but roughly equivalently active in DF1/MCT1 cells (this might reflect differences in MCT1 expression levels in the two systems). This ambiguity led us to leave out a discussion of this point in the original paper – however we recognize that it perhaps should have been discussed and we have mentioned it in the revision.

*3) The experiments showing that HERV-T envelope expression can block infection are convincing, but I have some concerns with the evidence that the hsaHTenv ORF was maintained due to selective pressure. The evolution simulation results are interesting, but not very informative without knowing the number of HERV-T proviruses in the genome, e.g. if there were 20 HERV-T env ORFs that integrated during that time period, on average we would expect one ORF to be maintained by chance, according to this simulation. The analysis is further complicated by the presence of older proviruses that would have had a smaller, but non-zero probability of having their env ORFs preserved to the present day by chance. It would be helpful to have information on dN/dS ratios for these sequences – if this gene has been exapted some level of purifying selection would be expected. If its only function was to protect against a long-extinct virus the selective pressures would have been relaxed since then, which might obscure the signs of selection somewhat, but it seems likely that at least some signature of selection at the residue level would remain.*

While we concede that simulated evolution by itself, is not the strongest evidence that hsaHTenv was preserved to provide function, we do believe that our simulation experiment is informative in the context of other evidence and arguments. We have modified the text to include such considerations, and strengthen the case that hsaHTenv indeed preserved:

A) We agree with the reviewers that (dN/dS) analyses could add support to the notion that hsaHTenv has been preserved. Due to the small number (three) of orthologous integrations for this locus (Human, Gorilla and Orangutan), we performed pairwise comparisons between these sequences and included the results in Table 1. Briefly, we found signs of purifying selection (dN/dS < 1) between the human and orangutan sequences that, unlike the gorilla sequence, show a preserved env ORF. This result is consistent with the notion the purifying selection acted on this gene following its insertion into ancestral hominids. Note that a possible need to inactivate fusogenenicity would mitigate against the detection of purifying selection at this locus.

B) While it is true that ~1:20 env genes of the same age would be expected to be preserved by chance, and the presence of older proviruses slightly complicates the interpretation of our simulations, this argument makes the assumption that there is no penalty (fitness cost) associated with expression of an intact env gene – which we think is unlikely, given that ancHTenv is fusogenic this is mentioned in the revised Discussion.

C) We note the quite striking difference in the number of inactivating mutations in the Gag and Pol genes of the orthologous provirus as compared to Env. This observation, we think, significantly strengthens the argument that hsaHTenv was selectively preserved.

D) We also note that the hsaHTenv and orthologous orangutan and Gorilla sequence are mutated in a similar way – i.e. at the SU-TM boundary, preserving receptor binding, but inactivating fusogenicity.

*4) For reasons that are unclear to us, this paper was submitted as a Short Report, although it contains much more than the "single series" of experiments specified for this format. The result is that some results that could be included in the results are relegated to supplemental, and both the Introduction and Discussion are severely constrained. There is a reasonably rich body of literature on exaptation of endogenous viruses that prevent infection by exogenous retroviruses that is barely mentioned here. These include not only the ALV and MLV case briefly mentioned in the Discussion, which, like HERV-T encode blocking env proteins, but also MMTV and JSRV which use very different mechanisms. Also, worth discussing is the possible contribution of the furin cleavage mutations or the C-terminal deletion in exaptation. The reader is shortchanged by the brevity enforced by the format.*

We concede the original manuscript was a little terse, and that some data was relegated to supplementary figures so as to fit into the short report format. We have modified the manuscript to the standard manuscript format.